# The Therapeutic Effects of EFNB2-Fc in a Cell Model of Kawasaki Disease

**DOI:** 10.3390/ph16040500

**Published:** 2023-03-28

**Authors:** Yijing Tao, Wei Wang, Yihua Jin, Min Wang, Jiawen Xu, Yujia Wang, Fangqi Gong

**Affiliations:** Department of Cardiology, Children’s Hospital, Zhejiang University School of Medicine, National Clinical Research Center for Child Health, No. 3333 Binsheng Road, Hangzhou 310052, China

**Keywords:** Kawasaki disease, EphB4, EphrinB2-Fc, coronary artery endothelial cells

## Abstract

The EphrinB2/EphB4 signaling pathway involves the regulation of vascular morphogenesis and angiogenesis. However, little is known about EphrinB2/EphB4 in the pathogenesis of Kawasaki disease (KD) and coronary artery aneurysm formation. Hence, this study aimed to explore the role of EphrinB2/EphB4 and the potential therapeutic effect of EphrinB2-Fc in the coronary arterial endothelial injury of KD. The levels of EphB4 were compared between KD patients and healthy children. Human coronary artery endothelial cells (HCAECs) were stimulated with sera from acute KD patients to establish the KD cell model. The overexpression of EphB4 or treatment with EphrinB2-Fc was found to intervene in the cell model. The cell migration, angiogenesis, and proliferation ability were assessed, and the expression of inflammation-related factors was measured. Our study showed that EphB4 showed low expression in both KD patients and the cell model of KD. The EphB4 protein levels in the CECs of CAA+ KD patients were much lower than those in healthy children. EphrinB2-Fc treatment of KD sera-activated HCAECs suppressed cell proliferation, reduced the expression of inflammation-related factors (such as IL-6 and P-selectin), and elevated cell angiogenesis ability. The results reveal that EphrinB2-Fc has a protective function in endothelial cells and has promising clinical applications for protecting vascular endothelium in patients with KD.

## 1. Introduction

Kawasaki disease (KD) is an acute, self-limited febrile illness characterized by systemic inflammation in all medium-sized arteries. Cardiovascular manifestations can be prominent during the acute KD episode and are the leading cause of long-term morbidity and mortality. In the past, although KD has surpassed rheumatic fever to become the most common cause of acquired heart disease in children, public understanding of KD and research into its treatment were inadequate. Nowadays, the pandemic caused by the novel coronavirus COVID-19 has become an urgent public event of global concern. Several reports suggest that children infected with severe acute respiratory syndrome coronavirus 2 (SARS-CoV-2) may develop a type of multisystem vasculitis, and this multisystem inflammation has a lot in common with KD [1,2]. Therefore, some have warned that KD may be one of the important complications of SARS-CoV-2 in children [3,4]. Thus, the COVID-19 outbreak brought more attention to KD, and more people started to think about what kind of disease it is and how to cure it. Although the application of intravenous immunoglobulin (IVIG) decreases the incidence of coronary artery aneurysms, about 4% of patients are still not cured [5].

The cause and pathogenesis of KD remain unknown. A common hypothesis is that the pathogenesis of Kawasaki disease may be related to a superantigenic toxin mechanism. That is, innate immune pathogen-associated molecular patterns (PAMPs) from microorganisms activate pro-inflammatory signals in innate immune cells and blood vessel cells, leading to abnormal activation of the body’s immune system. Therefore, superantigens can activate T cells at a very low concentration, thus secreting a large number of cytokines, leading to immune disorders. In addition, a large number of damage-related molecular patterns (DAMPs) generated by cell death and oxidative stress in circulating blood have multipotent effects on platelets, monocytes, neutrophils, T cells, endothelial cells, and vascular smooth muscle cells. In response to these stimuli, monocytes and neutrophils from peripheral blood are recruited into stimulated vascular cells [6]. The disruption of endothelial cellular homeostasis may have a crucial role in the pathogenesis of early KD vasculitis and may potentially be responsible for the development of coronary artery outcomes [7].

The receptor tyrosine kinase EphB4 and its ligand EphrinB2 (EFNB2) play a critical role in vasculature development [8]. They regulate critical aspects of endothelial cell behavior in a cell–cell contact-dependent fashion. The EphrinB2/EphB4 signaling pathway involves the regulation of sprouting angiogenesis and vascular morphogenesis [9]. The loss of EphB4 impairs the mechanical properties and transport function of endothelial cells, resulting in capillary ruptures and metabolic alterations [10]. Thus, we hypothesize that EphB4/EFNB2 may play a vital role in the pathogenesis of KD vasculitis and artery aneurysms. Since the first therapeutic Fc-fusion protein was introduced for the treatment of AIDS by Capon et al. [11], a dozen Fc-fusion proteins have been approved by the FDA [12]. Among these medications, Etanercept, approved for the treatment of rheumatoid arthritis, was the most successful therapeutic Fc-fusion protein [13]. Currently, many novel Fc-fusion proteins are undergoing pre-clinical and clinical development [14]. Recombinant human EphrinB2 Fc chimera (EFNB2-Fc) is an Fc-fusion protein, and it has been used in multiple studies to stimulate EphB4 signaling [15,16,17]; it may provide new directions for the treatment of KD.

To verify the above hypothesis, the following studies were performed. First, the level of EphB4 in circulating endothelial cells (CECs) from KD patients was quantitatively determined. Second, human coronary artery endothelial cells (HCAECs) were stimulated with sera from acute KD patients to establish the KD cell model. In the corresponding KD cell model, EphB4 levels were knocked down with siRNA, and the migration and angiogenesis capacity of vascular endothelial cells under the condition of low expression of EphB4 were detected. EphB4 was also overexpressed in the KD cell model to analyze the role of EphB4 in cell proliferation and inflammation. Finally, the therapeutic effects of EFNB2-Fc treatment on proliferation, angiogenesis, and inflammation in KD cell models were evaluated.

## 2. Results

### 2.1. EphB4 Protein Level of CECs Were Lower in the KD Patients

Flow cytometric analysis was performed to analyze the level of EphB4 protein in circulating endothelial cells (CECs). CECs are derived from the damaged vascular endothelium and reflect the damage degree of vascular endothelial cells. CECs were identified as cells lacking CD45 expression, positive for CD34, positive for CD144, and negative for CD133 (CD45−/CD34+/CD144+/CD133−) in this study (Figure 1A). In total, 26 patients with acute KD and 20 healthy children were enrolled in this study. The clinical characteristics and laboratory findings are listed in Appendix A. Among the KD patients enrolled, the mean percentage level of CECs in PBMC was 0.2795 ± 0.0761. That was significantly higher than the level in the HC group (0.0205 ± 0.0046; *p* < 0.0001) (Figure 1B). The mean number of CECs per million PBMC in the HC group was 205.2 ± 46.19, while it was 2795 ± 760.8 in the KD group (Figure 1C). To explore whether EphB4 is related to coronary artery aneurysm (CAA) in KD, according to the occurrence of CAA or not during the course, KD patients were divided into CAA+ KD and CAA- KD groups, which were, respectively, compared with the HC group. The result showed that the level of EphB4 protein in CECs was lower in the CAA- KD group than in the HC group (*n* = 18 vs. *n* = 20, *p* = 0.0497) and was much lower in the CAA+ KD group than in the HC group (*n* = 8 vs. *n* = 20, *p* = 0.0027) (Figure 1D). The mean EphB4 level of the KD group as a whole was also lower than that of the HC group (*n* = 26 vs. *n* = 20, *p* = 0.0028) (Figure 1E).

In summary, the protein level of EphB4 of CECs was also lower in the CAA- KD group than in the HC group and was even lower in the CAA+ KD group.

### 2.2. The Expression of EphB4 and EFNB2 in the Cell Model of KD Was Decreased

Human coronary artery endothelial cells (HCAECs) were stimulated with 15% sera from acute KD patients for 6 h to establish the KD cell model. Healthy children’s sera were used to stimulate HCAEC as a control group in the cell model to exclude any potential interference from human serum. In the cell model of KD, the expression of EphB4 and the ligand EFNB2 were detected by RT-qPCR and Western blot. The protein levels of both EphB4 and EFNB2 decreased (EphB4, *p* < 0.01; EFNB2, *p* < 0.01) (Figure 2A,B). The RNA level of EphB4 decreased, whereas no significant difference in EFNB2 was found (EphB4, *p* < 0.05; EFNB2, *p* = 0.1202) (Figure 2C).

### 2.3. EphB4 Inhibition Promoted Cell Migration and Attenuated Cell Angiogenesis Ability

Endothelial cell migration and proliferation are key processes in the repair and aggravation of vascular injury. To determine the functional significance of low EphB4 expression in KD, EphB4 was knocked down with specific siRNA targeting EphB4. Meanwhile, siCtrl was transfected into cells as a control group to eliminate the interference of transfection reagent and RNA on cells. After 24 h of knockdown, a linear scratch was made in HCAECs, and the medium was changed with basic ECM. The result indicated that EphB4 knockdown in HCAECs increased cell migration within 6 h (*n* = 4, paired Student’s *t*-test, *p* < 0.01) (Figure 3A,B).

In vitro angiogenesis assay with HCAECs revealed that KD sera treatment reduced tube formation ability (number of nodes, *p* < 0.05; number of junctions, *p* < 0.05; number of meshes, *p* < 0.05; total tube length, *p* < 0.01) (Figure 3C). HCAECs were collected after EphB4 knockdown for 24 h for in vitro angiogenesis assay. The result showed that EphB4 inhibition also reduced tube formation ability in HCAECs (number of nodes, *p* < 0.05; number of junctions, *p* < 0.05; number of meshes, *p* = 0.2165; total tube length, *p* = 0.1161) (Figure 3E), which is consistent with KD sera-treated HCAECs.

Knockdown of EphB4 was verified by Western blot and the protein level decreased (*p* < 0.05) (Figure 3D). Herein, pathways associated with angiogenesis, such as ERK1/2 and STAT1, and inflammation-related factor IL-6 were detected. The protein levels of EphB4, p-ERK, and p-STAT1 were lower when Ephb4 was knocked down (p-ERK, *p* < 0.05; p-STAT1, *p* < 0.05), whereas the protein levels of IL-6 showed no significant differences between the siEphB4 group and siCtrl groups (*p* = 0.1961) (Figure 3F). Cell proliferation did not show significant changes after knockdown (Figure 3G).

In summary, EphB4 knockdown in HCAECs increased cell migration and decreased cell angiogenesis.

### 2.4. Overexpression of EphB4 Suppressed KD-Induced High Cell Viability and Inflammation

In the cell model of KD, cell proliferation increased (24 h, *p* = 0.0746; 48 h, *p* = 0.0105) (Figure 4A). To investigate the effect of reversing low EphB4 expression in KD, HCAECs were transfected with EphB4 plasmid (Ad-EphB4) using Lipofectamine^TM^ 3000 (Invitrogen) according to the manufacturer’s protocol. Empty plasmids were transfected into cell models as a control group for our experiment of overexpressing EphB4, which aimed to intervene in the KD serum cell model. Since the KD serum cell model has reduced levels of EphB4, our goal was to reverse this effect by overexpressing EphB4. After 24 h of transfection, the medium was changed with 15% KD sera and continued to incubate for 6 or 24 h. Cell proliferation and expression of inflammation-related factors were compared between the two groups. Overexpression of EphB4 was verified by Western blot, and the protein level increased (*p* < 0.05) (Figure 4B). The result showed that cell proliferation in the Ad-EphB4 group was lower than in the Ad-null group (KD 6 h, *p* < 0.01; KD 24 h, *p* < 0.01) (Figure 4C). The protein levels of IL6 and SELP decreased in the Ad-EphB4 group (IL-6, *p* < 0.05; SELP, *p* < 0.05) (Figure 4D). That is, overexpression of EphB4 decreased cell proliferation and expression of inflammation-related factors in the corresponding KD cell model.

### 2.5. EFNB2-Fc Suppressed KD-Induced High Cell Viability and Inflammation and Elevated Cell Angiogenesis Ability

Recombinant human EphB2 Fc chimera (EFNB2-FC) or recombinant human IgG1 Fc (IgG Fc) was coated to the 6- and 96-well plates overnight before seeding the HCAECs. An IgG Fc intervention cell model was employed as a control group to eliminate any potential interference from IgG Fc in the EFNB2-Fc fusion protein on the cell model. This method was based on the study of Salvucci et al. [18]. After the HCAECs were incubated for 24 h, the medium was changed with 15% KD sera and then incubated for 6 h. Crosslinked EFNB2-Fc could potently activate the EphB4 protein. Cell proliferation and angiogenesis were compared between the two groups. The result showed that cell proliferation in the EFNB2-Fc group was lower than in the IgG Fc group (without KD, *p* < 0.01; KD 6 h, *p* < 0.0001) (Figure 5A). HCAECs in 6-well plates were collected for in vitro angiogenesis assay after incubating in 15% KD for 6 h. The result showed that EFNB2-Fc treatment elevated the tube formation ability in HCAECs (number of nodes, *p* < 0.05; number of junctions, *p* < 0.05; number of meshes, *p* = 0.0879; total tube length, *p* = 0.1465) (Figure 5B).

Angiogenesis-associated pathway ERK1/2 and some inflammation-related factors were detected. The protein levels of SELP, IL-6, and p-ERK were decreased (SELP, *p* < 0.05; IL-6, *p* < 0.001; p-ERK, *p* < 0.01) (Figure 5C,D).

In summary, EFNB2-Fc decreased KD-induced cell proliferation and inflammation and increased the angiogenesis of HCAECs stimulated with KD sera.

## 3. Discussion

COVID-19 is an infectious disease with substantial cardiovascular implications. In the case that humans may have to coexist with SARS-CoV-2, the incidence of KD or Kawasaki-like vasculitis is increasing yearly, and coronary artery involvement has been reported in 24% of patients [19]. Therefore, the treatment of KD or multisystem inflammatory syndrome in children is critical. Currently available treatments, other than IVIG, mainly target the acute phase symptoms of vasculitis, but the ultimate protective effect on coronary arteries is uncertain. In our laboratory, we conducted prior studies that revealed the potential cytoprotective effects of CsA. Specifically, we found that CsA has the ability to improve endothelial cell homeostasis by acting on the Ca^2+^/NFAT signaling pathway [20]. Additionally, our investigations showed that inhibiting KCa3.1 in macrophages can suppress the inflammatory reaction in a mouse cell model of KD [21]. In fact, some studies have demonstrated that EFNB2’s reverse signaling capability can contribute to the suppression of NFAT expression [22]. However, the intricacies of the relationship between EFNB2/EphB4 and KD have not yet been fully understood, thus necessitating further investigation and research. Therapeutic Fc fusion proteins have been used in other clinical diseases, so EFNB2-Fc has the potential for clinical application. Our research focuses on the therapeutic effect of EFNB2-Fc in protecting endothelial cells and preventing coronary aneurysms.

The main finding of this study is the downregulation of EphB4 in KD patients and in the cell model of KD. EphB4 protein levels in CECs from CAA+ KD patients were much lower than in healthy children, which shows a close link between EphB4 and coronary lesion in KD. Knockdown of EphB4 in HCAEC promoted cell migration and attenuated cell angiogenesis ability, which is consistent with the cell model of KD. Overexpression of EphB4 or EFNB2-Fc treatment suppressed KD-induced high cell proliferation. Furthermore, EFNB2-Fc treatment of KD sera-activated HCAECs decreased the expression of inflammation-related factors (such as IL-6 and P-selectin) and increased the angiogenesis ability of the cells. EFNB2-Fc shows a therapeutic effect in protecting endothelial cells in KD. These findings provide novel insights for potential KD treatment strategies.

Studies have shown that vascular endothelial dysfunction persists for a long time in KD patients and is closely related to long-term cardiac harm [23]. The coronary artery lesion has always been one of the research hotspots on the pathogenesis of KD. CECs in peripheral blood are derived from the damaged vascular endothelium, and their elevation can reflect the damage degree of vascular endothelial cells [24,25]. Our study confirmed that CECs were elevated in KD patients. The result indicated that EphB4 expression in CECs of KD patients was reduced, especially in CAA+ patients. Many studies have demonstrated that EphB4 is required for the integrity and homeostasis of cardiac vasculature [10,26]. Therefore, the downregulation of EphB4 may contribute to the dysfunction of endothelial cells in KD.

In this study, the cell model from a previous study [20] of our laboratory was used, that is, using sera of KD patients to stimulate HCAEC. In the cell model of KD, the protein levels of both EphB4 and EFNB2 decreased. The RNA level of EphB4 decreased, whereas no significant difference in EFNB2 was found. Our assumption is that the interplay between cytokines and EFNB2 may happen swiftly, causing mRNA to potentially degrade before protein levels reach their peak. To study the role of the reduction in EphB4, we knocked down EphB4 with specific siRNA targeting EphB4. The result indicated that EphB4 inhibition promoted cell migration. Previous studies have shown that plasma from patients with acute KD markedly enhanced the migration of endothelial cells [27]. We used to believe that cytokines act as stimulators of endothelial cell migration. Here, we found that EphB4 inhibition also contributes to cell migration. In addition, the knockdown of EphB4 attenuated cell angiogenesis ability, similar to HCAECs stimulated with KD sera. One previous report also indicated that impaired tube formation was closely linked to coronary aneurysms [28]. The decrease in tube formation may be associated with the severity of the vascular injury. ERK is a member of the MAPK pathway, and activation of ERK has been connected with vascular angiogenesis. Previous studies have suggested that inhibition of p-ERK inhibited endothelial tube formation [29,30]. On the contrary, decreased STAT-1 activation promotes the formation of vascular structures [31]. Knockdown of EphB4 led to the inhibition of both p-ERK and p-STAT1, but the ultimate effect on tube formation is reduction. A previous study showed that EphB4 relieved the release of proinflammatory factors [32]. In our study, the knockdown of EphB4 showed a tendency to increase inflammation-related factors, such as IL-6, but did not bring a significant difference. Cell proliferation did not show significant changes after knockdown. It is presumed that this is because the knockdown only resulted in a reduction in gene expression, leaving residual proteins that could sustain cell function. This suggests that even minimal levels of EphB4 have a crucial impact on cell proliferation.

To intervene in the cell model of KD, two therapeutic methods were used in our study. The mRNA and protein levels of EphB4 were decreased in HCAECs treated with KD sera, so we overexpressed EphB4 in the cell model. The result showed that EphB4 overexpression suppressed KD-induced cell proliferation. IL-6 has been implicated in the pathogenesis of a variety of systemic inflammatory diseases. Laboratory data showed that serum IL-6 levels were elevated in children with KD [33]. P-selectin, expressed in endothelial cells and platelets, is a specific molecular marker of platelet activation and is the initiating factor that mediates the interaction between platelets, leukocytes, and endothelial cells. Previous reports revealed that serum soluble P-selectin levels from patients in the subacute phase of KD were significantly higher than those with other febrile diseases or healthy children [34]. Our result showed that EphB4 overexpression reduced the expression of inflammation-related factors (such as IL-6 and P-selectin). In line with this, Kwan et al. [35] found that EphB4 was activated by EFNB2 to significantly inhibit IL-6 expression, and Ge et al. [32] found that EphB4 can alleviate the release of pro-inflammatory factors. In the second method, the cell model was treated with coated EFNB2-Fc. Previous studies showed that treatment with EFNB2-Fc would activate EphB4 signaling and ensure normal and functional angiogenesis [36,37]. In our study, although the activation of ERK1/2 was decreased, the result showed that treating with EFNB2-Fc enhanced the angiogenesis of HCAECs stimulated with KD sera. There might be other pathways involving the regulation of angiogenesis under the treatment of EFNB2-Fc. In addition, EFNB2-Fc could also suppress cell proliferation in the cell model of KD and reduce the expression of inflammation-related factors, such as IL-6 and P-selectin. In summary, EFNB2-Fc suppressed KD-induced cell proliferation and inflammation and enhanced the angiogenesis of HCAECs stimulated with KD sera. The results indicate that EFNB2-Fc had a protective function on endothelial cells. EFNB2-Fc has been applied in mice to increase vascularization in some studies [38]. The application of therapeutic Fc-fusion proteins in the clinic is in progress. Thereby, EFNB2-Fc has promising clinical applications to protect vascular endothelial cells in patients with KD.

In conclusion, EphB4 is reduced in the CECs of KD patients and the KD cell model constructed by serum stimulation, which may be linked to coronary lesions in KD. Knocking down EphB4 promoted migration and reduced angiogenesis. Overexpressing EphB4 or using EFNB2-Fc reduced cell proliferation in the KD cell model. EFNB2-Fc treatment decreased inflammation and increased angiogenesis, potentially protecting endothelial cells in KD. Our findings suggest a close relationship between EphB4 downregulation and endothelial cell lesions, and EphB4 may represent a new marker for the clinical prediction of coronary artery aneurysms in KD patients. EFNB2-FC, a drug targeting EphB4, suppressed KD-induced high cell viability and inflammation and enhanced cell angiogenesis ability, which may provide new directions for the treatment of KD.

## 4. Materials and Methods

### 4.1. Human Subjects

We collected peripheral blood samples from 26 patients with acute KD and 20 healthy children under well-child checking at Children’s Hospital, Zhejiang University School of medicine, in 2021. The whole blood was used to obtain peripheral blood mononuclear cells. The sera of another 60 patients with acute KD and 20 healthy children were collected for cell model construction and control treatment. To eliminate any potential bias, we combined sera from several CAA- patients and utilized them for cell modeling. KD was diagnosed according to the diagnosis criteria established by the American Heart Association [5]. All serum samples were filtered and stored at −80 °C until usage.

The study was conducted in accordance with the Declaration of Helsinki. All parents of the participants provided informed consent to participate in this study and approval was provided by the Institutional Review Board of Children’s Hospital affiliated to Zhejiang University school of medicine on 16 July 2019 (IRB number: 2019-IRB-073).

### 4.2. Flow Cytometry Analysis

Peripheral blood samples (2 mL each) were collected from the KD patients before IVIG and healthy children. PBMC were extracted using Ficoll-Paque PLUS (GE Healthcare Life Science) according to the manufacturer’s protocol. CECs were evaluated using a panel of antibodies: BV421-conjugated CD34 (BD Pharmingen, San Jose, CA, USA), PerCP/Cyanine5.5-conjugated CD45 (BioLegend, San Diego, CA, USA), phycoerythrin (PE)-conjugated CD133 (BioLegend), and Alexa Fluor^®^ 647-conjugated CD144 (BioLegend). Alexa Fluor 488-conjugated EphB4 (R&D Systems, Minneapolis, MN, USA) was used to analyze the level of EphB4. CECs were identified as cells lacking CD45 expression, positive for CD34, positive for CD144, and negative for CD133 (CD45−/CD34+/CD144+/CD133−).

PBMC were blocked with TruStain FcX™ (BioLegend) for 10 min and then incubated with CD34, CD45, CD133, CD144, and Alexa Fluor 488-conjugated EphB4 (R&D Systems, MN, USA) for 15 min in the dark. Fluorescence intensity was examined by BD FACSLyric (BD) and analyzed by flow cytometry software FlowJoTM 10 (BD).

### 4.3. Cell Culture and Preparation

Primary human coronary artery endothelial cells (HCAECs, obtained from ScienCell, CA, USA) were maintained in an endothelial culture medium (ECM). The medium contained 5% fetal bovine serum (FBS) with 1% endothelial cell growth supplement (ECGS) and 1% penicillin/streptomycin (P/S). Cells were cultured in a humidified atmosphere containing a 5% CO_2_ incubator at 37 °C. HCAECs were used for experiments between the third and fifth passage and were seeded into 96-, 24-, or 6-well microplates for assay when the culture reached 80% to 90% confluency. Healthy controls (HCs) and KD controls comprised HCAECs at 90% confluence incubated for 6 h in an endothelial cell basal medium-2 (basic ECM) with 15% healthy children or KD patient sera, respectively.

### 4.4. Reverse Transcription-Real Time Quantitative Polymerase Chain Reaction (RT-qPCR)

Total RNA from HCAECs was extracted using TRIzol^®^ (Invitrogen, Waltham, MA, USA) and then reverse-transcribed with PrimeScript^TM^ RT Master Mix (Takara, Kusatsu, Japan). Real-time quantitative PCR analyses were performed. The PCR condition for the reactions was as follows: 2 min at 95 °C and then 40 cycles of 5 s at 95 °C and 30 s at 60 °C. Forward and reverse primers are listed in Table 1. The relative mRNA level was calculated using the 2^(−△△Ct)^ method.

### 4.5. Western Blot Assay

The cells were harvested, and total proteins were extracted by RIPA lysis buffer containing protease and phosphatase inhibitors. The lysed solution was centrifuged at 12,000 rpm for 10 min at 4 °C. The protein concentrations were assessed by the BCA protein assay. The proteins were separated in 10% SDS-PAGE gel and transferred onto PVDF membranes (Millipore, Burlington, MA, USA). The membranes were blocked with 5% nonfat milk in TBST for 1 h at room temperature and incubated with the primary antibodies EFNB2 (1:1000), EphB4 (1:1000), SELP (1:2000), p-ERK (1:2000), p-STAT1 (1:1000), IL-6 (1:1000), and GAPDH (1:5000) at 4 °C overnight. The membranes were then incubated with corresponding horseradish peroxidase (HRP)-conjugated secondary antibodies (1:10,000) for 1 h at room temperature. The blots were interacted with chemiluminescence substrate (Thermo scientific, West Hills, CA, USA). Densitometry of bands was quantified by Quantity One software version 4.6.2 (Bio-Rad, Hercules, CA, USA).

### 4.6. Recombinant Human EphrinB2 Fc Chimera

Recombinant human EphrinB2 Fc chimera (EFNB2-FC, obtained from R&D) or recombinant human IgG1 Fc (IgG Fc, as a control for EFNB2-Fc, obtained from R&D) was coated at 2 μg/mL overnight at 4 °C. Then, we used PBS to wash the wells before seeding the cells. Following 24 h of incubation, the medium was changed to 15% KD sera in basic ECM for 6 h.

### 4.7. Cell Viability Assay

The cell viability was monitored using the cell counting kit 8 (CCK-8) assay. HCAECs were seeded in a 96-well plate at 3 × 10^3^/well density. The cells were grown overnight at 37 °C in humidified 5% CO_2_ and then treated with different sera, siRNA, or plasmid followed by specific incubation hours. After that, the medium was added with 10 μL CCK-8 (APExBIO) for 4 h. The optical density at a wavelength of 450 nm was measured (Bio Tek). The HCAEC were cultured in 5% ECM medium for the control wells, while simple medium was used for the blank wells. CCK reagent was added to the experimental, control, and blank wells, followed by measuring absorbance at 450 nm. To calculate the proliferation capacity, the ratio of (OD of experimental well—OD of blank well) to (OD of control well—OD of blank well) was determined. The methodology described in Figure 4A,C and Figure 5A all follows this approach.

### 4.8. In Vitro Angiogenesis Assay

After specific treatment, a total of 2 × 10^4^ cells per well were seeded onto the surface of the polymerized ECMatrix^TM^ (Millipore), incubated at 37 °C for 4 h, and photographed under an inverted light microscope at 40× magnification. The tube formation was analyzed by Image J software version 1.53a (National Institutes of Health, Bethesda, MD, USA).

### 4.9. Wound-Healing Assay

A scratch wound-healing assay was used to measure the cell migration rate. HCAECs were cultured in six-well plates and transfected with siEphB4 or siCtrl when the culture reached 80% to 90% confluency. A linear scratch was made using a sterile 200 μL tip and rinsed with D-PBS to remove the cell debris. Then, we changed the medium to basic ECM. The migration area was monitored at 0 and 6 h following wounding with an OLYMPUS CKX53 microscope and analyzed using Image J software, version 1.53. The cell relative migration rate (%) was defined as the ratio of the original area—the open area at a specific time to the original area.

### 4.10. Statistics

All the experiments were repeated at least three times, and the data are expressed as the mean ± SEM. All the data were analyzed using the GraphPad Prism Software version 8.0 (GraphPad Software Inc., Satiago, CA, USA). The statistical analysis involving two groups was performed using a two-tailed Student’s *t*-test. The comparison of more than two groups was carried out using one-way ANOVA. Values of *p* < 0.05 were considered indicative of significance.

## Figures and Tables

**Figure 1 pharmaceuticals-16-00500-f001:**
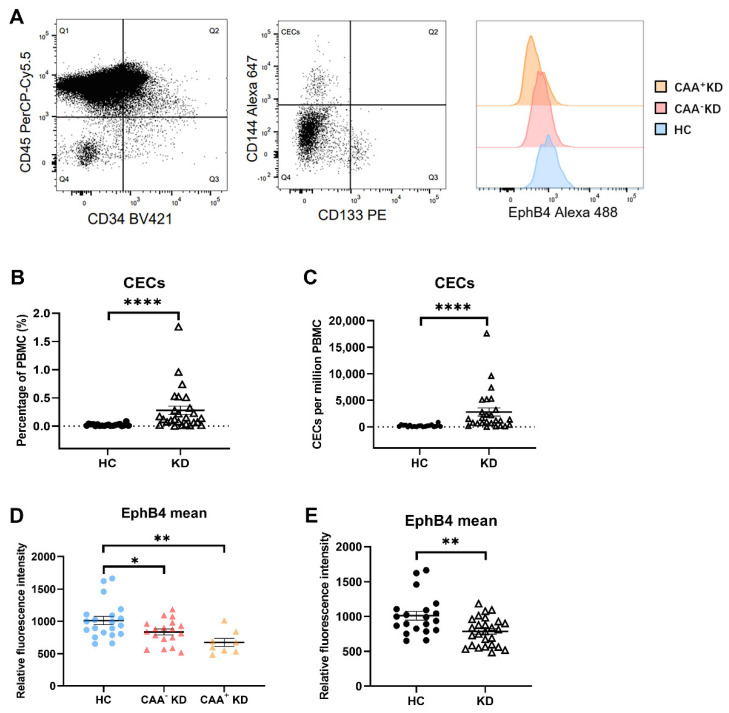
The expression of EphB4 was compared in KD patients and HCs. (**A**) Flow cytometric detection of CECs. The analysis gate (Q3) included CD34+ CD45- cells. The expression of CD133 and CD144 in the Q3 gate was detected. CECs were defined as CD34+CD45-CD133- and CD144+. The expression of EphB4 in CECs was detected. (**B**) Percentage level of CECs in PBMC was analyzed by FlowJo. (**C**) Numbers of CECs per million PBMC were compared between HCs and KD patients. (**D**) The mean fluorescent intensity of EphB4 in CECs of HCs, CAA- KD, and CAA+ KD was assayed. (**E**) The mean fluorescent intensity of EphB4 in CECs of HCs and KD patients was compared. The bar graphs are the mean ± SEM. * *p* < 0.05, ** *p* < 0.01, **** *p* < 0.0001 compared with the respective control group.

**Figure 2 pharmaceuticals-16-00500-f002:**
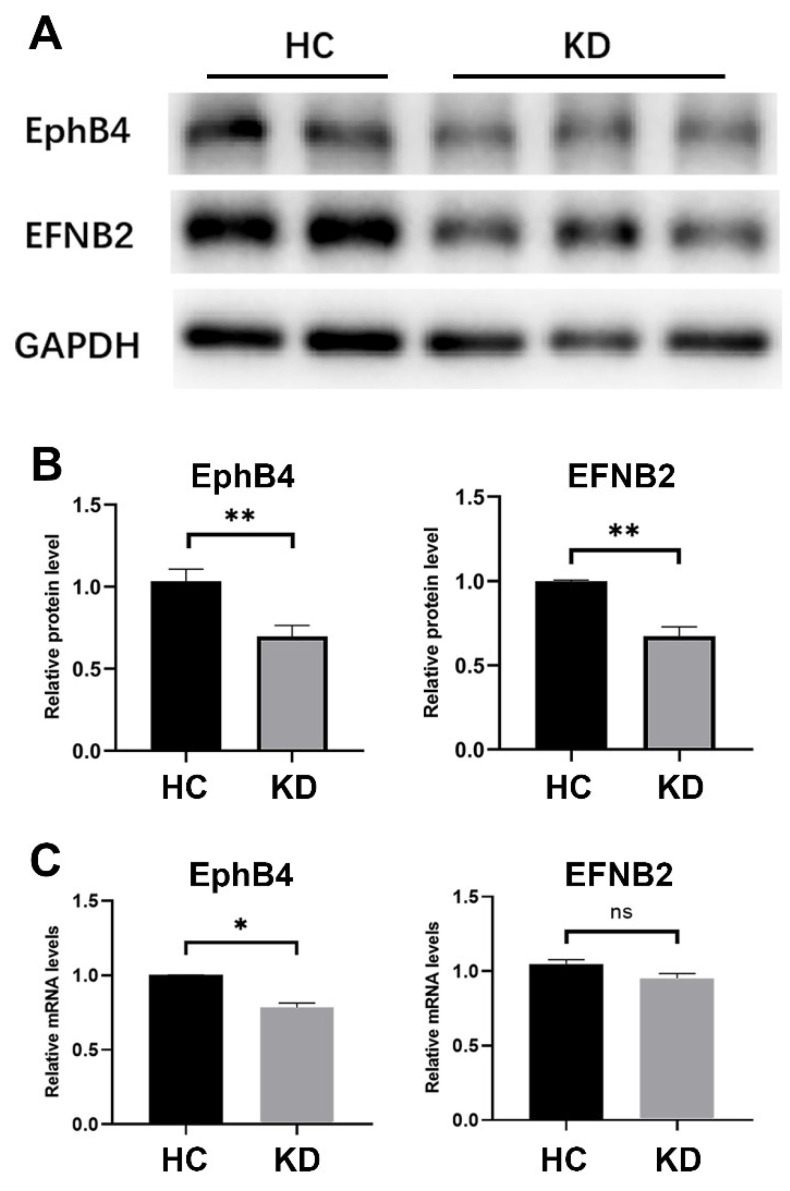
Expression of EphB4 and EFNB2 in the cell model of KD. HCAECs were incubated in a medium containing 15% sera from HCs or KD for 6 h. (**A**,**B**) Protein levels of EphB4 and EFNB2 were measured by Western blot. (**C**) The mRNA expressions of EphB4 and EFNB2 were measured by RT-qPCR. All results are expressed as the mean ± SEM. * *p* < 0.05, ** *p* < 0.01 vs. the control group.

**Figure 3 pharmaceuticals-16-00500-f003:**
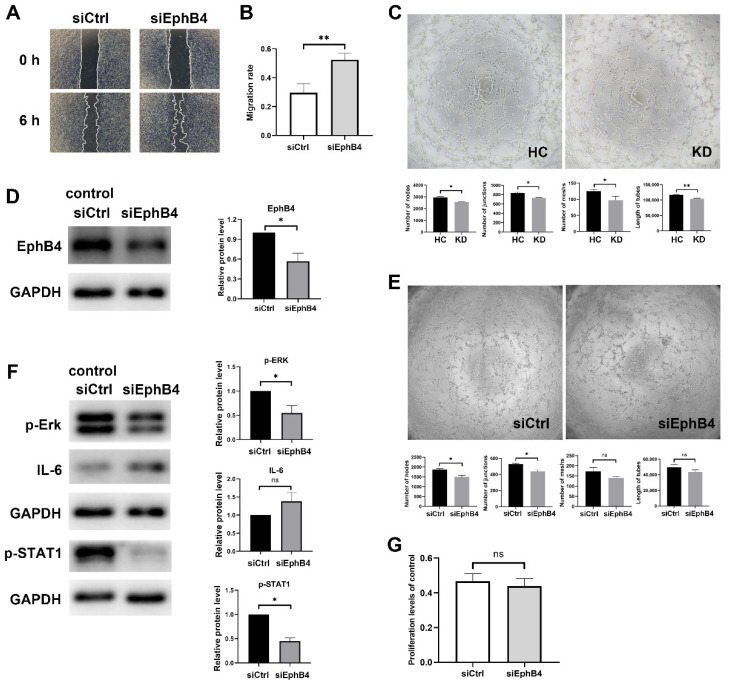
Effects of EphB4 inhibition on cell migration and angiogenesis ability. (**A**,**B**) HCAECs were transfected with siRNA for 24 h. The cell migration ability was measured by wound-healing assay and analyzed using Image J software. (**C**) HCAECs were pretreated with a medium containing 15% KD or HC sera for six hours and then were plated on Matrigel. (**D**) The protein level of EphB4 was detected in the siEphB4 group. (**E**) HCAECs were plated on Matrigel after transfected with siEphB4 or control siRNA for 24 h. (**F**) Effects of EphB4 inhibition on the protein expressions of p-ERK, IL-6 and p-STAT1 were analyzed by Western blot. Full-length blots are presented in Appendix A. (**G**) Cell proliferation did not show significant changes after knockdown. All results are expressed as the mean ± SEM. * *p* < 0.05, ** *p* < 0.01 vs. the control group.

**Figure 4 pharmaceuticals-16-00500-f004:**
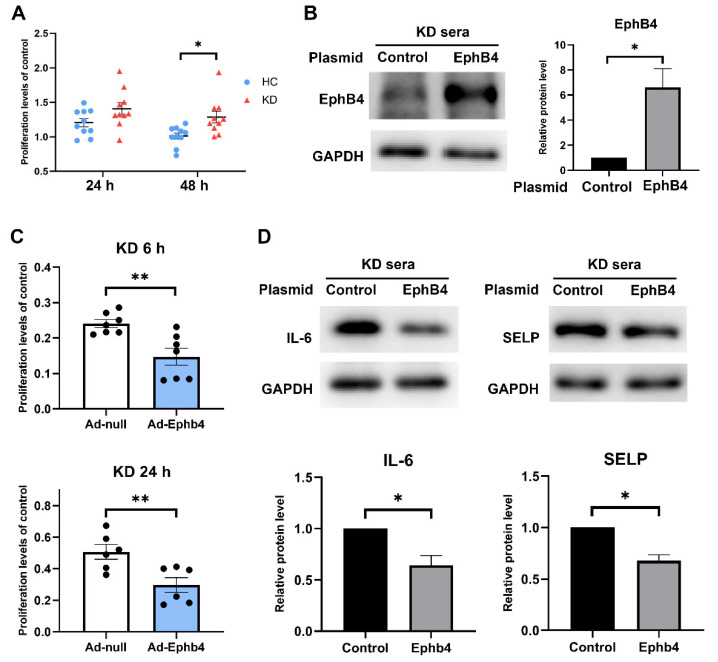
Overexpression of EphB4 suppressed KD-induced high cell viability and inflammation. (**A**) HCAECs were incubated in medium containing 15% sera from HC or KD. After 24 h and 48 h, cell proliferation was detected by CCK-8. (**B**) HCAECs were transfected with EphB4 or ctrl plasmid for 24 h and then incubated in 15% KD sera for 6 h. Then, the protein level of EphB4 was detected by Western blot. (**C**) HCAECs were transfected with plasmid for 24 h and then incubated in 15% KD sera for 6 or 24 h. The cell proliferation was measured using CCK-8 assay. (**D**) HCAECs were transfected with EphB4 or ctrl plasmid for 24 h and then incubated in 15% KD serum for 6 h. The protein levels of IL-6 and SELP were measured via Western blot. Full-length blots are presented in Appendix A. All data are the mean ± SEM. * *p* < 0.05, ** *p* < 0.01 vs. the control group.

**Figure 5 pharmaceuticals-16-00500-f005:**
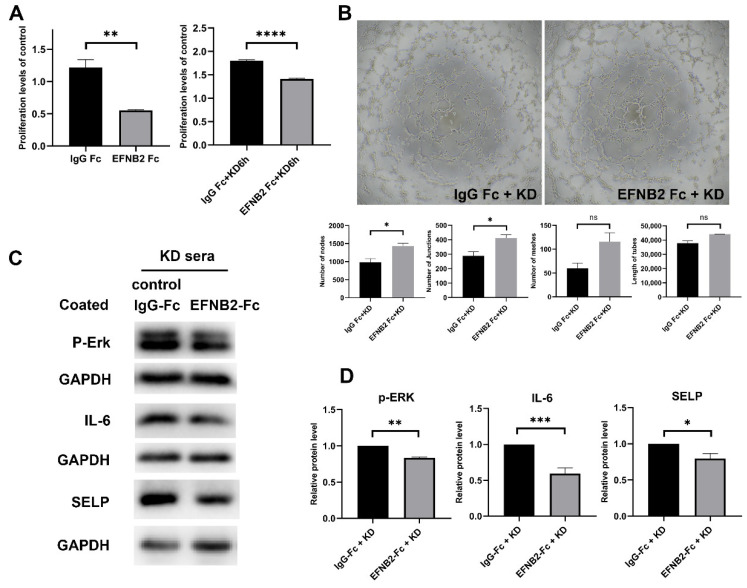
Effects of EFNB2-Fc on cell proliferation, angiogenesis ability, and inflammation. (**A**) HCAECs were seeded in plates precoated with EFNB2-Fc or IgG Fc and incubated for 24 h. The medium was then changed with 15% KD sera for six hours or not, and cell proliferation was detected by CCK-8. (**B**) HCAECs were seeded in plates precoated with EFNB2-Fc or IgG Fc, incubated for 24 h, changed medium with 15% KD sera for 6 h, and then plated on Matrigel. (**C**,**D**) HCAECs were seeded in plates precoated with EFNB2-Fc or IgG Fc, incubated for 24 h, and changed medium with 15% KD sera for 6 h. The protein levels of p-ERK, IL-6, and SELP were measured by Western blot. Full-length blots are presented in Appendix A. All data are the mean ± SEM. * *p* < 0.05, ** *p* < 0.01, *** *p* < 0.001 and **** *p* < 0.0001 vs. the control group.

**Table 1 pharmaceuticals-16-00500-t001:** RT-qPCR primers used in this study.

Gene	Forward Primer (5′→3′)	Reverse Primer (5′→3′)
EphB4	CCACCGGGAAGGTGAATGTC	CTGGGCGCACTTTTTGTAGAA
EFNB2	TATGCAGAACTGCGATTTCCAA	TGGGTATAGTACCAGTCCTTGTC
GAPDH	GGAGCGAGATCCCTCCAAAAT	GGCTGTTGTCATACTTCTCATGG

## Data Availability

Data are contained within the article and Appendix A.

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
