# Peer review of "The Therapeutic Effects of EFNB2-Fc in a Cell Model of Kawasaki Disease"

_pharmaceuticals, 2023, doi:10.3390/ph16040500_

Round 1
Reviewer 1 Report
where is the control in western blot figures.
Please relate with previous work
Resolution of figures is inferior
Reviewer 2 Report
Dear Authors,
the paper is scientifically extremely interesting and very nicely written. Considering the extensive results, I think it would be good to devote a separate chapter to the conclusion in order to make the main findings and innovations of this work clearer.
Reviewer 3 Report
Lines 65-69: authors describe clinical uses of Fc-fusion proteins but need to add a sentence regarding EFNB2-Fc as a fusion protein
Lines 278-280 : change soluble E selectin to P-selectin
Lines 302-305: please clarify peripheral blood samples in 26 KD patient vs sera : that whole blood was to obtain PBMC
Reviewer 4 Report
Dear Miss Chelariu
Please find included my comments on the paper “The therapeutic effects if EFNB2-Fc in a cell mocel of Kawasaki disease” by Tao Y
The paper is basically more a validation of their cell model and the exploration of the EphrinB2/EphB4 pathway based on observations then investigating therapeutic potential.
I have two major concerns with this paper: the lack of a mechanism. And technical: Controls using the serum of healthy volunteers is missing and should be included.
Main: Lack of mechanism
Why is chosen to measure for these inflammatory cytokines in the serum? IL-6 is increased in the serum of KD patients (see supplemental table). It is also increased in endothelial cells after KD administration. What is the rational of investigating IL-6? What happens when IL6 is neutralized in the serum? Is IL6 the driving factor?
Main: lack of controls
Figure 4 and 5. Only KD serum is investigated. Show also the results with serum from HC.
Figure 4A and line 164: “proliferation increased” compared to what? What is the control? What means proliferation levels: ratio? percentage? How is this calculated?
Same for 4C and 5A
Minor
1.Serum of KD patients is added to HCAECs. Is this serum for CAA+ or CAA- patients? If yes, please include. If not, include the rational not to separate?
2.Figure 2. Protein levels of EFNB2 are significant, mRNA levels are not. Please explain why that would be.
3.Figure 3. Show proliferation as well.
Reviewer 5 Report
Review for the
Manuscript ID: pharmaceuticals-2246647
Type of manuscript: Article
Title: The therapeutic effects of EFNB2-Fc in a cell model of Kawasaki disease
Authors: Yijing Tao, Wei Wang, Yihua Jin, Min Wang, Jiawen Xu, Yujia Wang *, Fangqi Gong *
Submitted to section: Pharmacology,
https://www.mdpi.com/journal/pharmaceuticals/sections/pharm-pharmacology
EPH and Ephrins in Pathogenesis and as Drug Target
https://www.mdpi.com/journal/pharmaceuticals/special_issues/EPH_Ephrins
In this study authors aims to explore the role of EphrinB2/EphB4 and the potential therapeutic effect of EphrinB2-Fc in the coronary arterial endothelial injury of Kawasaki disease (KD). The authors showed that EphB4 was low-expressed in both KD patients and the cell model of KD. The EphB4 protein levels in CECs of CAA+ KD patients were much lower than in healthy children. EphrinB2-Fc treatment of KD sera-activated HCAECs suppressed cell proliferation, reduced expression of inflammation-related factors and elevated cell angiogenesis ability. The result reveals that EphrinB2-Fc has a protective function on endothelial cells and has promising clinical applications to protect vascular endothelium in patients with KD.
The manuscript is well prepared and clearly written, good in technical quality and only needs some minor corrections and improvements, listedin PDF attachment.
After all listed minor corrections will be done, the current manuscript can be accepted for publication.

Round 2
Reviewer 1 Report
None
Reviewer 4 Report
accept in present form